# AN EMPIRICAL STUDY OF A PRUNING MECHANISM

## ABSTRACT

Many methods aim to prune neural network to the maximum extent. However, there are few studies that investigate the pruning mechanism. In this work, we empirically investigate a standard framework for network pruning: *pretraining large network and then pruning and retraining it*. The framework has been commonly used based on heuristics, i.e., finding a good minima with a large network (pretraining phase) and retaining it with careful pruning and retraining (pruning and retraining phase). For the pretraining phase, the reason for which the large network is required to achieve good performance is examined. We hypothesize that this might come from the network relying on only a portion of its weights when trained from scratch. This way of weight utilization is referred to as *imbalanced utility*. The measures for weight utility and utility imbalance are proposed. We investigate the cause of the utility imbalance and the characteristics of the weight utility. For the pruning and retraining phase, whether the pruned-and-retrained network benefits from the pretrained network is examined. We visualize the accuracy surface of the pretrained, pruned, and retrained networks and investigate the relation between them. The validation accuracy is also interpreted in association with the surface.

## 1 INTRODUCTION

Deep learning is currently one of the most powerful machine learning methods. It requires neural network to train, which usually takes a few to hundreds times more weights than training data (He et al., 2016; Zagoruyko & Komodakis, 2016; Huang et al., 2017; Karen & Andrew, 2015). Usually, in common regimes, a greater number of weights leads to better performance (Zagoruyko & Komodakis, 2016). However, paradoxically, neural networks are also compressible. Many of the recent pruning methods aim to maximally compress the networks (Han et al., 2015; Liu et al., 2017; He et al., 2019; You et al., 2019), however, there are few works that investigate why and how the pruning mechanism works (Frankle et al., 2019; Elesedy et al., 2020).

In this work, we empirically investigate a standard framework for network pruning: *pretraining a large network and then pruning and retraining it*. The framework has been commonly used based on heuristics, i.e. finding a good minima with a larger network and retaining it with careful pruning and retraining (Han et al., 2015; Liu et al., 2017). We investigate the heuristic in two parts, i.e., one for the pretraining phase and the other for the pruning and retraining phase.

For the pretraining phase, the reason for training the large network to obtain a good minima is investigated. Since the neural network is generally compressible, the pretrained large network can be pruned to a smaller one. However, a network with the same number of weights as that of the pruned network cannot achieve similar performance when trained from scratch (Frankle & Carbin, 2018). We conjecture that this comes from the networks not utilizing all of their weights. Thus we hypothesize: if trained from scratch, there is a utility imbalance among the weights in neural network. For investigation, the measures for the weight utility and the utility imbalance are proposed. Thereafter, the cause of the utility imbalance and the characteristics of the weight utility in various conditions are examined.

For the pruning and retraining phase, we verify the heuristic that once a good minima is obtained with the large network, it can be retained by careful pruning and retraining (Han et al., 2015; Renda et al., 2020). Our investigation is based on the loss surface visualization on a two dimensional plane formed by three points in the weight space where each point represents the pretrained network, the

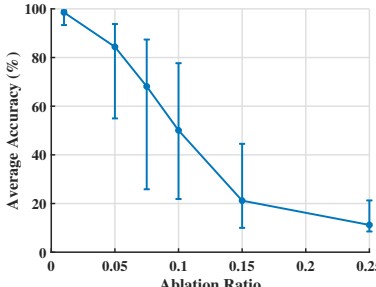 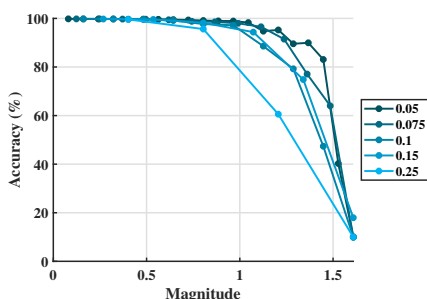

Figure 1: The examples of the imbalance in weight utilization. (left) We trained a network following the procedure for $1\times$ network in Section 2.2.1. The weights were randomly ablated from the network and the training accuracy was measured for 500 times. The average accuracy is plotted with the minimum and maximum values in the bar. Note that the accuracy difference is about 60% when the ablation ratio is 0.1. This implies the utility imbalance among the weights. (right) The training accuracy with respect to the ablation regarding the magnitude of the weights. Each legend indicates the ablation ratio. Note that the training accuracy drops only when the weights with the large magnitude is ablated. The utility of the weight is biased with regard to the magnitude.

pruned network, and the pruned and retrained network. We examine (1) the dynamics of the network on the loss surface throughout pruning and (2) the validation accuracy of the networks over varying pruning methods and retraining methods.

**Contributions.**

- The utility imbalance among the weights increases during optimization.

- The neural networks utilize the weights in proportion to their size.

- If a pretrained network is carefully pruned and retrained, then the pruned-and-retrained network shares the same loss basin with the pretrained network.

## 2    WEIGHT UTILITY ANALYSIS FOR THE PRETRAIN MECHANISM

Then why do we have to train a large network and then prune to a smaller one? Why not just train the smaller one to get the performance we need? Why is it difficult? The investigation about the questions starts with a hypothesis: let $N_{large}$ be a large network that does not utilize all of its weights, and thus can be easily compressed into a smaller network $N_{pruned}$ with minimal loss change. And let $N_{small}$ be a network trained from scratch, whose number of weights is comparable to that of $N_{pruned}$, which is sufficient to achieve a similar level of loss to those of $N_{large}$ or $N_{pruned}$. However, $N_{small}$ generally performs worse, because $N_{small}$ *does not utilize all of its weights either*. Therefore, we hypothesize that the neural network does not utilize all of its weights when trained from scratch in general. And we refer to the phenomenon which the neural network utilize the weights unevenly as *utility imbalance*. Thus,

**Main Hypothesis.** If trained from scratch, there is utility imbalance among the weights in a neural network.

And we empirically measure the utility of weights as:

**Definition 1** (Utility measure). Let $W$ be a set of total weights in a network $N$, $W_s$ be a subset of $W$, and $X$ be a dataset. Suppose $f_W(x)$ and $f_{W \setminus W_s}(x)$ are probability mass functions resulting from a softmax layer, where $x \sim X$ is an input and $f_{W \setminus W_s}(x)$ is obtained by zeroing out the weights in $W_s$. Then, the utility of $W_s$ can be measured as $U(W_s) = \underset{x \sim X}{\mathbb{E}} \left[ d_{KL} \left( f_W(x), f_{W \setminus W_s}(x) \right) \right]$, where $d_{KL}$ is KL-divergence.

For reference, the way of the measurement, i.e., network ablation, was similarly done in (Casper et al., 2019; 2020; Meyes et al., 2019; Cheney et al., 2017). We also define the utility imbalance as:

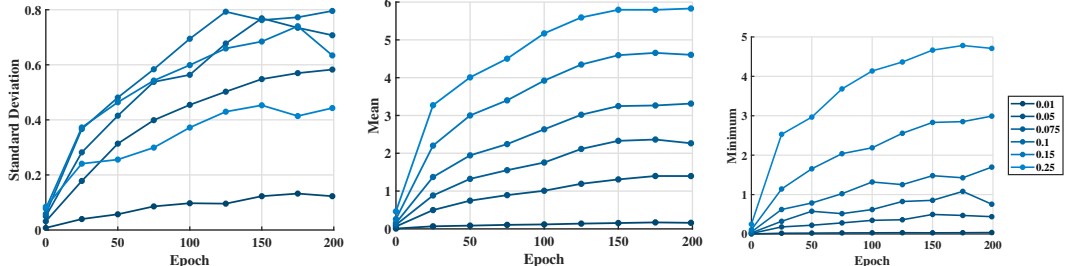

Figure 2: The weight utility characteristics during optimization. Each legend indicates the ablation ratio. (left) The measured value of the utility imbalance, i.e., the standard deviation of the weight utility of the sample set. (middle) The average weight utility of the sample set. (right) The minimum weight utility in the sample set. The corresponding training accuracy is in Figure 7

.

**Definition 2** (Utility imbalance). For $W_i \subset W$ and $W_j \subset W$, we say there is $\delta$ utility imbalance between the subsets of weights when $|U(W_i) - U(W_j)| > \delta$.

Where we empirically measure the utility imbalance by:

**Definition 3** (Utility imbalance measure). For a set of the randomly drawn subsets of $W$, i.e., $\{W_i\}_{i=1}^{N}, W_i \subset W$, which we refer to as a *sample set*, we empirically measure the utility imbalance of the set by the standard deviation of $\{U(W_i)\}_{i=1}^{N}$.

We empirically show that the utility imbalance among the weights exists in Figure 1. Hereafter, the cause of the utility imbalance and the characteristics of the weight utility are discussed in Section 2.1 and 2.2, respectively.

## 2.1 THE CAUSE OF THE UTILITY IMBALANCE

In this section, we investigate the cause of the utility imbalance among the weights. The cause driven from initialization and optimization is discussed in Section 2.1.1 and 2.1.2, respectively.

### 2.1.1 FROM INITIALIZATION

The utility imbalance among the weights can be given by initialization. The Lottery Ticket Hypothesis (Frankle & Carbin, 2018) showed that there is a fortuitous subset of weights at initialization, namely winning tickets, which achieves the commensurate accuracy with the original network within the commensurate training time when trained in isolation. Since they uncovered the winning tickets by pruning the weights with the smallest magnitude after pretraining the whole network, we can say that the weights in the winning ticket are the ones that become the largest when trained as a whole. And since the weights with the larger magnitude tend to be more utilized than the others (Han et al., 2015), it can be inferred that the weights in the winning ticket are the well-initialized ones that can become the most utilized ones after trained as a whole. However, the winning tickets become less effective when the subnetworks are trained to be far apart from the initialization, e.g., trained with a larger learning rate or using ResNet (He et al., 2016) or Batch Normalization (Ioffe & Szegedy, 2015; Hoffer et al., 2018), where the gradients flowing through the network are larger (Frankle & Carbin, 2018).

### 2.1.2 DURING OPTIMIZATION

The utility imbalance among the weights can also be intensified during optimization. Although Frankle & Carbin (2018) conjectured that SGD may seek out and train the winning tickets, they did not give further evidence. Here, we conduct an experiment to measure the utility imbalance during training.

We classified CIFAR10 dataset (Krizhevsky, 2009) using a vanilla convolutional neural network (vanilla CNN) (the architecture of the network is specified in Section A.2.1), trained by SGD with zero momentum and weight decay. The network was trained for 200 epochs with initial learning rate

of 0.1, which is reduced by $10\times$ at 50% and 75% of the training epochs. When we measured the utility of a subset (refer to Definition 1), the weights in the subset were zeroed out for the exclusion, and KL-divergence was used to measure the distance between the probability outputs of the original network and the ablated network. For each subset, the distance was measured by averaging over the total dataset. To measure the utility statistics, e.g., utility imbalance, we used a set of 500 randomly selected subsets ($N = 500$ in Definition 3) which we refer to as *sample set*. We only used the training data in the experiment for two reasons: (1) it is more straightforward to interpret and (2) in our experiments, the validation accuracy was higher when the training loss was lower, thus we regarded that the probability output of the training data represented the validation accuracy to some degree.

From the left figure in Figure 2, we can see that the utility imbalance, i.e., the standard deviation of the utility of the subsets in the sample set, increases during training. Additionally, the mean of the utility increases (Figure 2, middle), which implies that the network is utilizing the weights more effectively and is becoming sensitive to the ablation as training proceeds. Because the utility given the amount of ablation ($|\delta|$) can be interpreted as $|\frac{\Delta f(W+\delta)}{\Delta \delta}|$, it is also inferred that the loss surface is becoming sharper. It corresponds with the previous work (refer to Ghorbani et al., 2019, Figure 3), where the scale of the eigenvalues of the Hessian with respect to the weights grows significantly over training. The increase in the average utility can be the reason for the increase in the utility imbalance – if statistically analyzed, the standard deviation of data grows larger when the data is scaled to be larger; or if intuitively interpreted, the network outputs with respect to the different ablation sets differ to a greater extent as the loss surface becomes sharper. Moreover, we can see that the minimum of the utility in the sample set also increases (Figure 2, right), which infers that each of the weights is struggling to be utilized more, rather than SGD purposely differ the utility among the weights. This still holds when we trained the network much longer, i.e., 5000 epochs, where we put the result in Figure 11. We also conducted the experiments with the most popular architecture for classification, i.e., ResNet20 (He et al., 2016) and Batch Normalization (Ioffe & Szegedy, 2015); and more advanced optimization techniques, i.e., momentum and weight decay. Please refer to Figure 12 and Figure 13 for the results. The results also hold for the above statements.

## 2.2 THE FEATURES OF THE UTILITY IMBALANCE

In this section, we investigate how neural networks utilize weights under different conditions. We show the characteristics of the weight utilization in different-sized networks in Section 2.2.1, and in pruned networks in Section 2.2.2.

### 2.2.1 UTILITY IMBALANCE AND NETWORK SIZE

Here, we investigate how the different-sized networks utilize their weights. For a fair comparison, instead of adjusting the width of the layers, i.e., the number of output channels of the layers, we randomly pruned the network at initialization and re-scaled the variance of the remaining weights according to the original initialization method, i.e. He initialization (He et al., 2015). This was to control the number of feature maps and thus avoid any architectural bias. We compared ResNet20 (He et al., 2016) with Batch Normalization (Ioffe & Szegedy, 2015) and its sampled subnetworks whose number of weights are $0.5\times$ and $0.25\times$ of that of the original network. The network was trained by SGD with momentum ($\alpha = 0.9$) and weight decay ($\lambda = 10^{-4}$). Other conditions are the same as the experiment in Section 2.1.2. We performed the ablation experiments under two different settings: where the number of ablated weights are (1) proportional to the number of weights in each network (which is notated as Ablation Ratio in Figure 3) and (2) the absolute number of weights that are equally applied to the networks (Ablation Number in Figure 3). Each scale of the ablation number in Figure 3 is a proportion to the number of weights in the $1\times$ network.

From Figure 3, we observe that no matter how large the network is, there is utility imbalance among the weights. Moreover, the three networks show the similar responses when the weights are ablated in proportion to the network size (Figure 3 top), but quite different when ablated by the same number (Figure 3 bottom). This is quite remarkable in that the network utilizes the number of weights proportional to its size rather than the absolute number of weights required to map a certain function. Casper et al. mentioned this way of weight utilization at initialization (refer to Casper et al., 2020, Appendix B). Thus, it is inferred that the tendency endowed from initialization remains to the end of training.

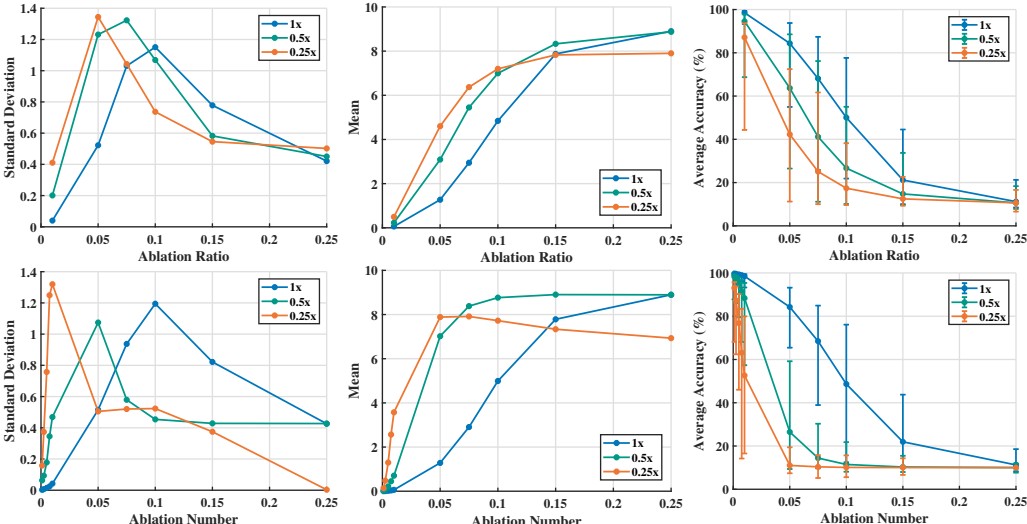

Figure 3: The characteristics of the weight utility of the networks with the various sizes. Each legend indicates the size of the network. (top) The characteristics when the weights are ablated by ratio. (bottom) The characteristics when the weights are ablated by the absolute number. Note that the characteristics of the networks are similar when ablated by ratio.

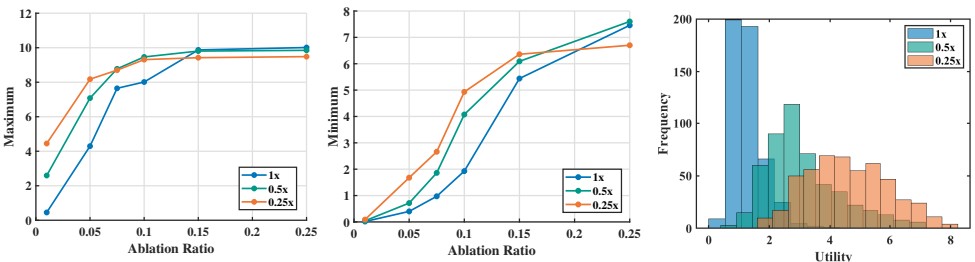

Figure 4: The other characteristics of the weight utilization. Each legend indicates the size of the network. (left) The maximum weight utility in the sample set. (middle) The minimum weight utility in the sample set. (right) The histogram of the values of the weight utility when the ablation ratio is 0.05. Please refer to the histograms with other ratios in Figure 14.

By taking a closer look, even when compared by ratio where the similar tendency is shown, the actual number of weights still matters to a lesser degree. For example, the maximum weight utility among the random weight sets (Figure 4, left) is generally higher in the smaller network. This implies that the most utilized weights in the smaller network are more capable than those in the larger one, even if more weights are considered for the larger network. This is a counterexample for the conjecture in the Appendix in Frankle & Carbin (2018), which says the larger network shows superior performance because it has better winning tickets. If the larger network had a better ticket with the same amount (ratio) of weights, its most utilized weights should have been more critical than those of the smaller one. Another interesting result comes from the average weight utilization (Figure 3, top middle). It is shown that the output of the larger network differs from that of the original network by the same level by the larger ablation ratio. This implies that the larger network relies on the larger portion of the weights. In addition, the larger network tends to less utilize the weights when ablated by the same ratio if the network has the ability to classify. This is in line with the findings of Casper et al., where the larger network was regarded to be more robust to the ablation (Casper et al., 2019). In summary, the larger network uses the less-utilized weights by the larger portion. We conjecture this may be a reason for the larger network showing the better performance. The tendency is also consistent with the maximum and the minimum of the utility, and is confirmed by the histogram of the utility values in Figure 4.

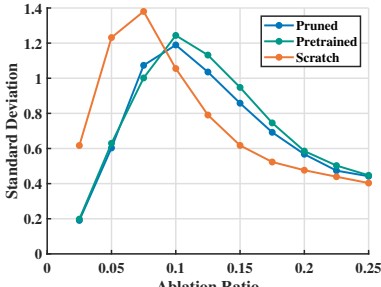 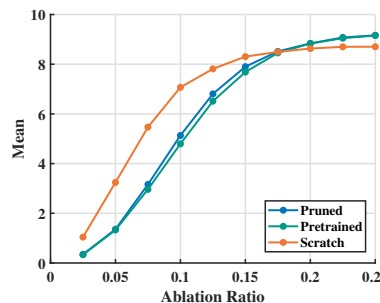

Figure 5: The characteristics of the weight utility in the pruned-and-retrained network. The legends indicate the pruned-and-retrained network, the pretrained network and the small network, respectively. Note that the characteristics of the pruned-and-retrained network are similar to that of the pruned-and-retrained one, even though we used the large learning rate for retraining.

### 2.2.2 UTILITY IMBALANCE IN PRUNED NETWORK

Here, we investigate the utility characteristics in a pruned network. The network architecture and optimization scheme are the same as the experiments in Section 2.2.1. We used the one-shot global magnitude pruning with respect to the individual weights, and only the weights in the convolution layers were pruned for convenience. We retrained the pruned network for another 200 epochs with the same learning rate schedule used for the pretraining (Renda et al., 2020). For comparison, we trained a small network which is trained from scratch and has the same number of weights as that of the pruned network. Especially, we trained the small network for 400 epochs to match the total training epochs of the pruned network. The learning rate schedule was also the same for the pruned-and-retrained network, which is twice the repetition of that of the pretraining. The utility statistics were acquired as the same as the experiments in Section 2.1.2. For the pretrained network, only the weights with the larger magnitude than pruning threshold were considered for the subsets for the utility measure ($W_s$ in Definition 1). This was to control the weights of interest, since only the weights larger in magnitude than the pruning threshold remain in the pruned network. Figure 5 is the result for the pruning ratio 0.5, and Figure 15 is for 0.25 and 0.75.

In Figure 5, the pruned-and-retrained network exhibits the similar utility characteristics to those of the pretained network. It is surprising in that we used large learning rates for retraining the pruned network, which were large enough for the pretrained network to achieve 100% training accuracy from a random state. Although the statistics of the pruned-and-retrained network differ from those of the pretrained network as the pruning ratio increases (Figure 15), still the former resembles the latter, even when the pruning ratio is 0.75. From the results, we conjecture that the high performance of the pruned-and-retrained network comes from the weight utility characteristics endowed by the large pretrained network, e.g., relying on the larger portion of the less-utilized weights, as in Section 2.2.1.

## 3 LOSS SURFACE ANALYSIS OF THE PRUNE-RETRAIN MECHANISM

Canonical pruning methods begin with training a large neural network, followed by identifying and removing a portion of less important parameters for compressing the network, yet preserving the accuracy. Such pretrain-and-prune paradigm is motivated by an implicit assumption that: (1) a large network trained with SGD optimization has a higher chance of converging to a minima with less generalization gap, and (2) pruned and retrained network converges to a minima with a close relation to that of the pretrained network. In this regard, we investigate the relation between the pretrained, pruned and retrained networks in this section.

Empirically, the generalization gap is known to be closely related to the geometry of the loss basin. Hochreiter & Schmidhuber (1995) first proposed that flat loss minima corresponds to less overfitting. Jiang et al. (2020) showed that measures based on flatness of minima such as sharpness (Keskar et al., 2017) are highly correlated with generalization performance. Also, Li et al. (2018) visualized the loss minima of a family of Wide-ResNets, and showed that larger networks converge to flatter

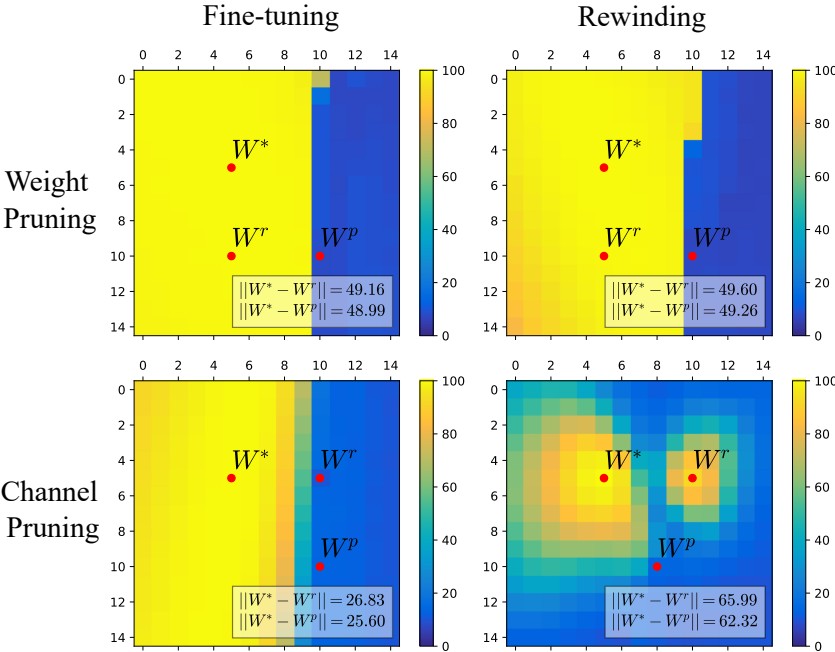

Figure 6: The visualization of the pretrained, pruned and retrained weights on the accuracy surface. The color map indicates training accuracy (%). Refer to Section A.5.2 for the trajectory change over the pruning cycles.

minima. This is consistent with the empirical observations that larger neural networks have smaller generalization gaps.

Likewise, we investigate the relation between the networks by visualizing the loss surface. To the best of our knowledge, there has been not been a clear verification on whether the assumptions for the heuristic hold. In order to verify such assumptions and shed some light on what happens through pruning, we visualize the trajectory of the pruning mechanism on a low-dimensional loss surface.

**Pruning and retraining methods.** The typical pruning algorithms prune and retrain the network iteratively to achieve high sparsity and avoid loss in accuracy. The iterative pruning cycle consists of two alternating phases: the pruning phase and the retraining phase. At the pruning phase, a portion of weights or channels are removed. The weight pruning methods aim to identify the importance of the individual weight of a neural network, while the channel pruning methods target the channels in a convolutional layer. Therefore, the channel pruning methods operates in a higher constraint, which often results in greater damage to the network. At the retraining phase, the pruned network is commonly fine-tuned using a small and fixed learning rate, which is usually the last learning rate used at the pretraining stage. In addition, Renda et al. (2020) has shown that rewinding the learning rate scheduling to an earlier time step can boost the validation accuracy of the retrained network.

**Visualization technique.** For a straightforward understanding, we plot the accuracy surface instead of the loss surface in this section. To visualize the points of interest, we follow the 2D planar loss visualization used in Garipov et al. (2018). We construct the 2D plane by affine combinations of three points in the weight space – the pretrained weight ($W^*$), the pruned weight in the last iterative cycle ($W^p$), and the final retrained weight ($W^r$).

**Experiment details.** For the experiments, CIFAR10 dataset (Krizhevsky, 2009) and ResNet20 (He et al., 2016) were used. At the pretraining stage, we use the same training hyperparameters and data augmentation strategy described in He et al. (2016); with SGD momentum $\alpha = 0.9$ and weight decay $\lambda = 10^{-4}$. The pretrained network was trained for 200 epochs, with an initial learning rate of 0.1 divided by 10 at 50% and 75% of the total training epochs. The number of retraining epochs was also 200 for each cycle. For comparison, the small networks with the same number of weights as that of the corresponding pruned networks were trained from scratch. The small networks are set by

Table 1: Validation accuracy comparison. "Prune" refers to the pruned-and-retrained network and "Small" refers to the small network trained from scratch. Note that the aggressive channel pruning with fine-tuning shows the low performance, where it often collapsed to 10% accuracy.

| Pruning method | Retraining method | Pruned | Small |
|---|---|---|---|
| weight pruning | fine-tuning | 90.870 | - |
| | rewinding | 91.897 | 89.290 |
| channel pruning | fine-tuning | 43.375 | - |
| | rewinding | 87.138 | 88.020 |

randomly pruning the large network at initial and rescaled the variance of the weights regarding the original initialization, i.e. (He et al., 2015). To be fair, we used the same number of training epochs and the same learning rate schedule as those of the pretrained-pruned-retrained network (Liu et al., 2018) for training the small network.

For the weight pruning, we used the magnitude pruning method (Han et al., 2015). And for the channel pruning, we used the criteria proposed in Network Slimming (Liu et al., 2017). To show the failure case of pruning, we applied the method much more aggressively (Section A.5.1). For each experiment, we performed the iterative pruning over five cycles. We removed 30% and 20% of weights per cycle for the weight and the channel pruning, respectively. Only the weights in the convolution layers were pruned for convenience. For retraining the pruned network, two learning rate scheme were used: fine-tuning and learning rate rewinding (Renda et al., 2020).

**Results.** In the case of the weight pruning, we consistently observe that the pretrained weight and the pruned-and-retrained weight are connected by the high-accuracy region on the accuracy surface (Figure 6, top). This indicates that the final weight is placed in the original loss basin (accuracy peak). Moreover, we find that the retrained weight vector remains in the same loss basin even with a aggressive retraining method, i.e., learning rate rewinding (Renda et al., 2020). Table 2 also shows that the pruned network has consistently better performance than the small network trained from scratch, which is in line with our observation in the visualization. It is implied that the pruned-and-retrained network is likely to remain in the flat loss basin reached at the pretraining stage. This observation is in agreement with the assumption that the pretrain-and-prune practice takes advantage of the large network to attain higher generalization performance.

On the other hand, in the case of the aggressive channel pruning, we observed that the two points are possibly in the separate loss basins when retrained by learning rate rewinding (Figure 6, bottom right). We conjecture this came from the aggressive channel pruning and the large learning rate of the learning rate rewinding. It is provable that the pruned weight shifted too much from the pretrained weight $W^*$, such that the retrained point $W^r$ locates in a different loss basin. To give further evidence, we compare the accuracy of the pruned-and-retrained network and that of the small network (Table 2). The result shows that the pruned-and-retrained network has a higher generalization gap than the small network, implying that it did not benefit from the pretrained network. Moreover, when using the fine-tuning method, the retrained network could not even recover from the pruning and remains in the low accuracy region (Figure 6, bottom left). Overall, for the aggressive pruning, the result does not seem to work as the initial assumptions.

## 4 DISCUSSION

A standard pruning framework, i.e., pretraining a large network and then pruning and retraining, was examined. To investigate the pretraining phase, we defined the measures for the weight utility and the imbalance of the weight utility. The cause of the weight imbalance and the characteristics of the weight utility were discussed. For the pruning and retraining phase, the relation between the pretrained network and the pruned and retrained network was investigated using the accuracy surface and the validation accuracy of the networks. The various conditions were examined to verify the heuristic.

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

## A    APPENDIX

### A.1    RELATED WORKS

**Pruning methods.** Network pruning attempts to compress pretrained deep neural networks by identifying and removing parameters that are estimated to be less important for the network. Unstructured pruning methods remove parameters at a fine-grained level by removing each of the weights individually. Han et al. (2015) proposed a method that assesses the importance of weights based on their norm, and also performs an additional fine-tuning step after pruning in order to compensate for the loss of parameters. Structured pruning on the other hand, removes parameters at a larger, filter or kernel level. Liu et al. (2017) came up with a method that prunes channels with small BatchNorm (Ioffe & Szegedy, 2015) scaling parameters, which have been pushed towards zero with L1 regularization. Recent methods (You et al., 2019; He et al., 2019) focus on pruning networks with skip-connections (He et al., 2016). He et al. (2019) prunes 53.4% of parameters with a slight accuracy drop. In this work, we want to reveal some of the reasons why deep neural networks can be successfully pruned.

**Loss surface.** A streamline of works focus on the loss surface of neural networks and its properties. Garipov et al. (2018) introduce the concept of mode connectivity; that simple linear pathways between local minima can be discovered. They demonstrate ways of finding and visualizing the loss landscape along such pathways on various modern architectures. (Evci et al., 2019; Frankle et al., 2019) analyze sparse neural networks in terms of mode connectivity. Frankle et al. (2019) discovered that the existence of linear paths is a key indicator of whether lottery ticket networks (Frankle & Carbin, 2018) can be discovered. On the other hand, Evci et al. (2019) found that linear paths

between a pruned network and a network randomly initialized from the same sparsity pattern are separated by high energy barriers. In this context, our paper brings new insight by exploiting mode connectivity to analyze the minima obtained before and after pruning.

**Lottery ticket hypothesis.** The lottery ticket hypothesis(Frankle & Carbin, 2018) explores the existence of a sparse subnetwork that can reach the commensurate performance of a dense network within the commensurate time when trained in isolation. The subnetwork is discovered by pruning a pretrained network based on the magnitude of its weights, whereas the surviving weights are restored back to their values at initialization. Such weights are called winning tickets, and a pruned network can be obtained by retraining them. The authors verified the existence of lottery tickets on the CIFAR-10 dataset with fully connected and convolutional networks. However, current methods of finding lottery tickets are only restricted to certain settings, i.e, learning rate warmup and iterative pruning. Moreover, the mechanism of the lottery tickets is still an open area of research. Elesedy et al. (2020) is a work that provides an insight to the mechanism with linear models, where they reformulated the iterative magnitude pruning as a process of feature alignment.

**Learning rate rewinding.** Existing literature on pruning adopt a retraining phase after pruning to compensate for the loss of the network. The most commonly used method is fine-tuning, which is a process of additional training with a small and fixed learning rate (Han et al., 2015; Liu et al., 2018), i.e., typically the last learning rate used for pretraining. Meanwhile, Renda et al. (2020) suggested a new learning rate scheme which is referred to as learning rate rewinding. Unlike fine-tuning, they propose to rewind the learning rate schedule to the earlier phase in pretaining. In addition, they explored whether the weight rewinding (Frankle & Carbin, 2018) is beneficial. On various datasets such as CIFAR-10, ImageNet, WMT-16 dataset, they verified that both of the rewinding methods were better than fine-tuning.

## A.2 Additional Material for section 2.1

### A.2.1 The architecture of the vanilla CNN

The vanilla CNN used for the experiment was composed of (conv3 3x16)-(conv3 16x16)-(conv3 16x16)-(maxpool)-(conv3 16x32)-(conv3 32x32)-(maxpool)(conv3 32x64)-(conv3 64x64)-(global average pooling)-(fc 64x10). A convolution layer with a spatial size of $3 \times 3$ is indicated as 'conv3'. And 'maxpool' and 'fc' indicate a max pooling layer and a fully-connected layer, respectively. There is a Rectified Linear Units (ReLU) (Glorot et al., 2011) after each convolution layer.

### A.2.2 The training accuracy for each ablation in Figure 2

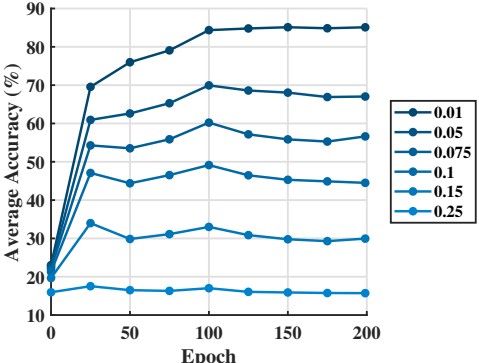

Figure 7: The training accuracy for each ablation in Figure 2. Each legend indicates the ablation ratio.

### A.2.3 THE DETAILED FIGURES FOR FIGURE 2

Since the data do not clearly show the tendency in the combined manner, we present the detailed figures for Figure 2 in Figure 8, 9, 10. The figures are ordered with respect to the corresponding ablation ratio of each datum, i.e., 0.01,0.05,0.075,0.1,0.15,0.32. In the figures, the statements in Section 2.1.2 generally hold.

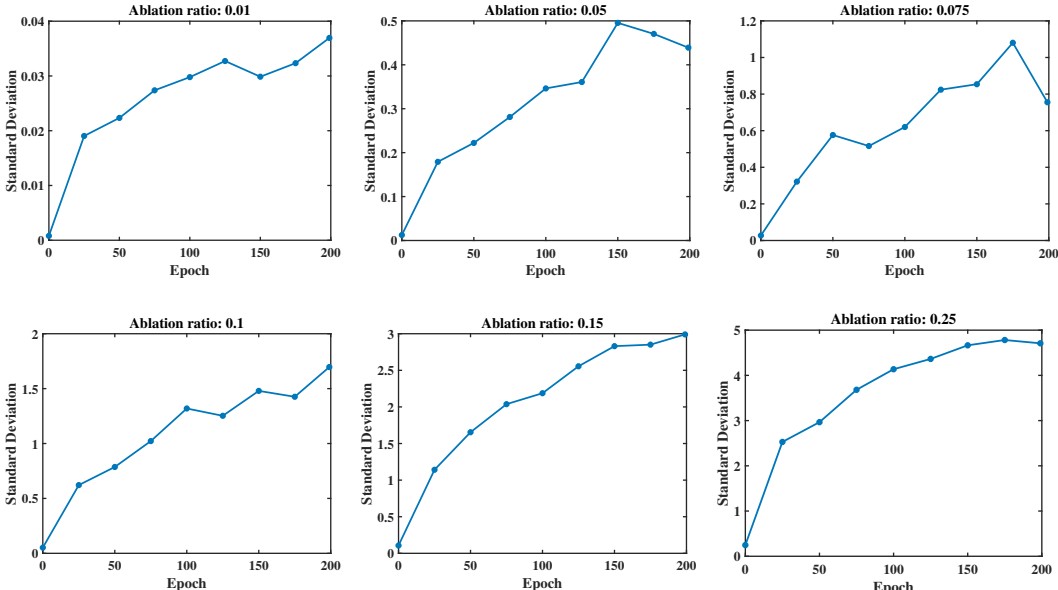

Figure 8: The standard deviation values of the weight utility in the sample set.

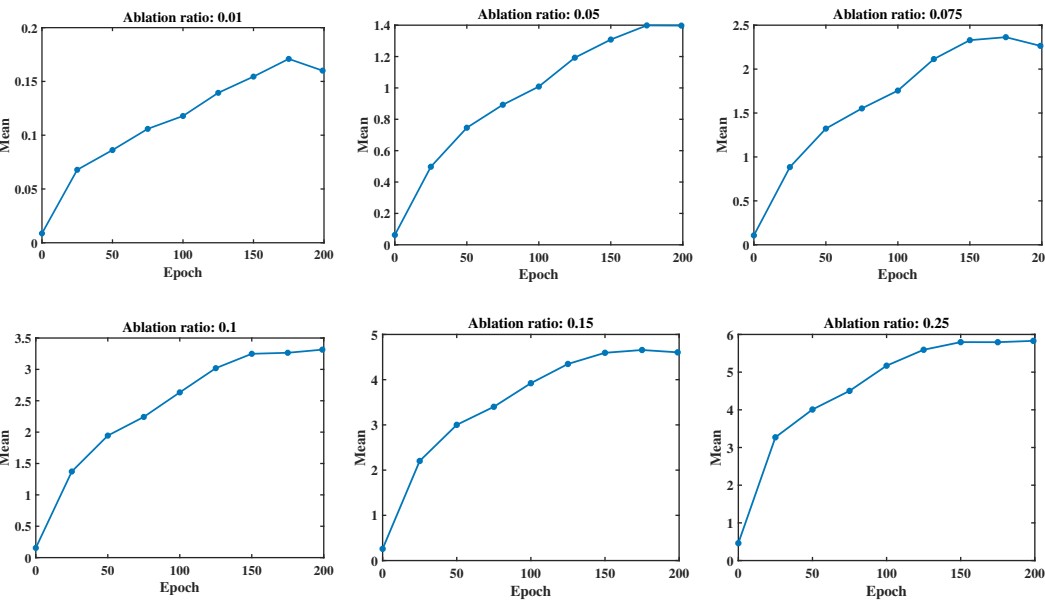

Figure 9: The average values of the weight utility in the sample set..

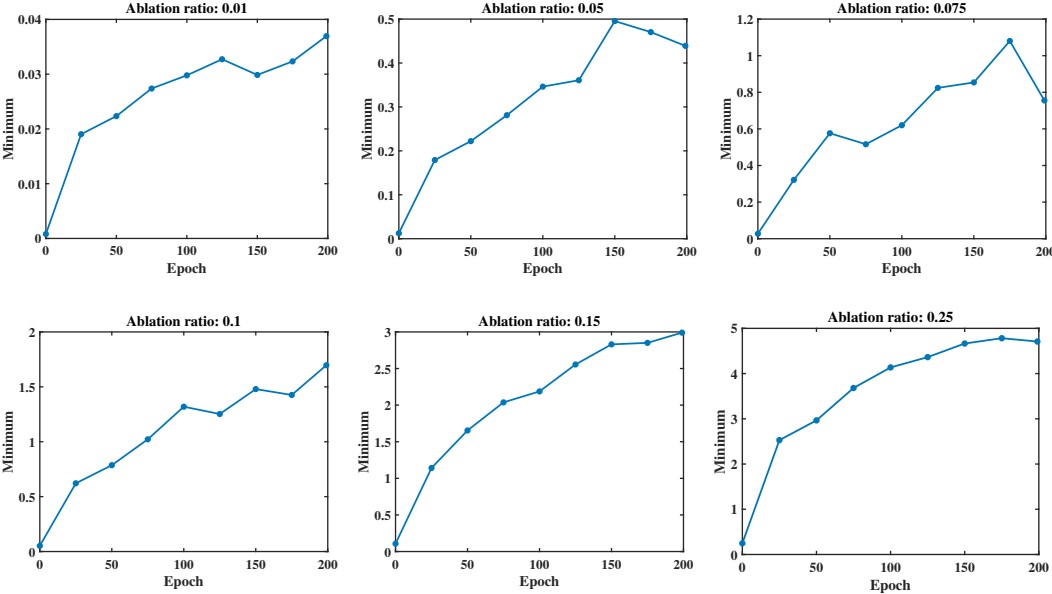

Figure 10: The minimum values of the weight utility in the sample set..

### A.2.4 The characteristics of the weight utility during optimization in varying conditions.

**The longer training epochs** With the same setting as the experiment in Section 2.1, we trained the network much longer than the conventions, i.e., 5000 epochs. It is about $25\times$ longer than the typical settings. The initial learning rate of 0.1 was reduced by $10\times$ at $50\%$ and $75\%$ of the total training epoch.

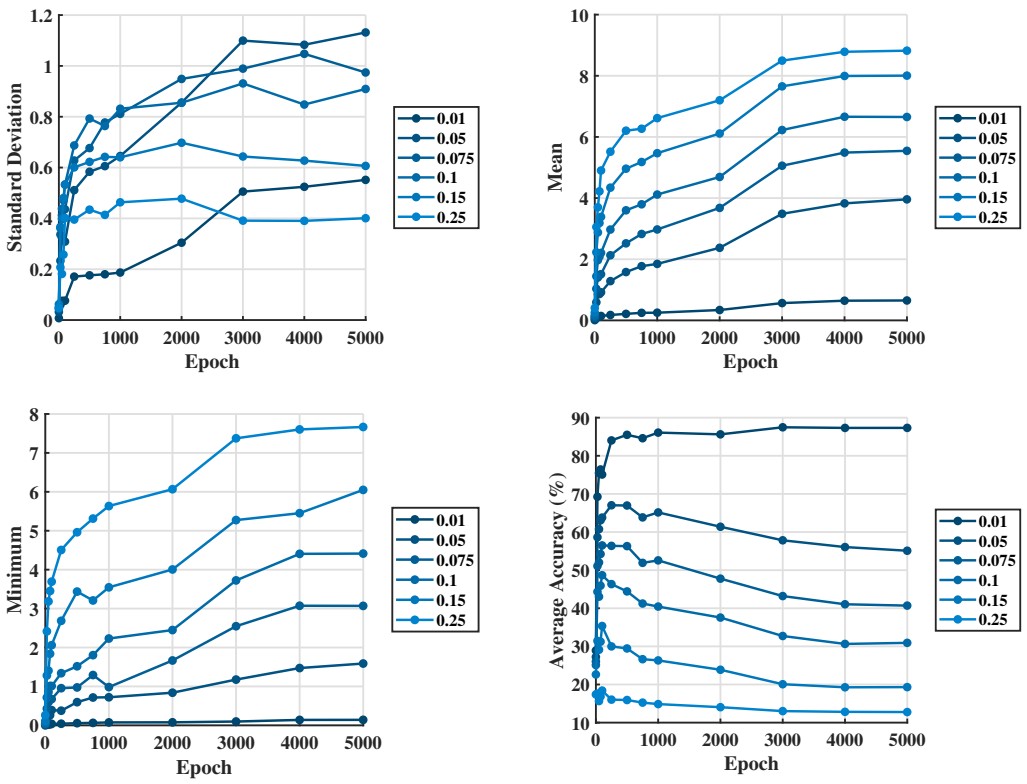

Figure 11: The utility characteristics of the vanilla network when trained for 500 epochs. Note the severe accuracy drop compared to that of Figure 7. This corresponds with the increase in the average weight utility, as the increase in the difference between the outputs of the ablated and the original network might cause the accuracy drop. In particular, the tendency is shown after the early phase of the training where the network undergoes an abrupt transition (approximately before the epoch 1000). Each legend indicates the ablation ratio.

**An experiment with the advanced settings** Here, the experiments was done with the advanced settings. We used ResNet20 (He et al., 2016) with Bathch Normalization (Ioffe & Szegedy, 2015), trained by SGD with momentum ($\alpha = 0.9$) and L2 weight decay ($\lambda = 10^{-4}$). Other settings are the same as the experiment in Section 2.1.

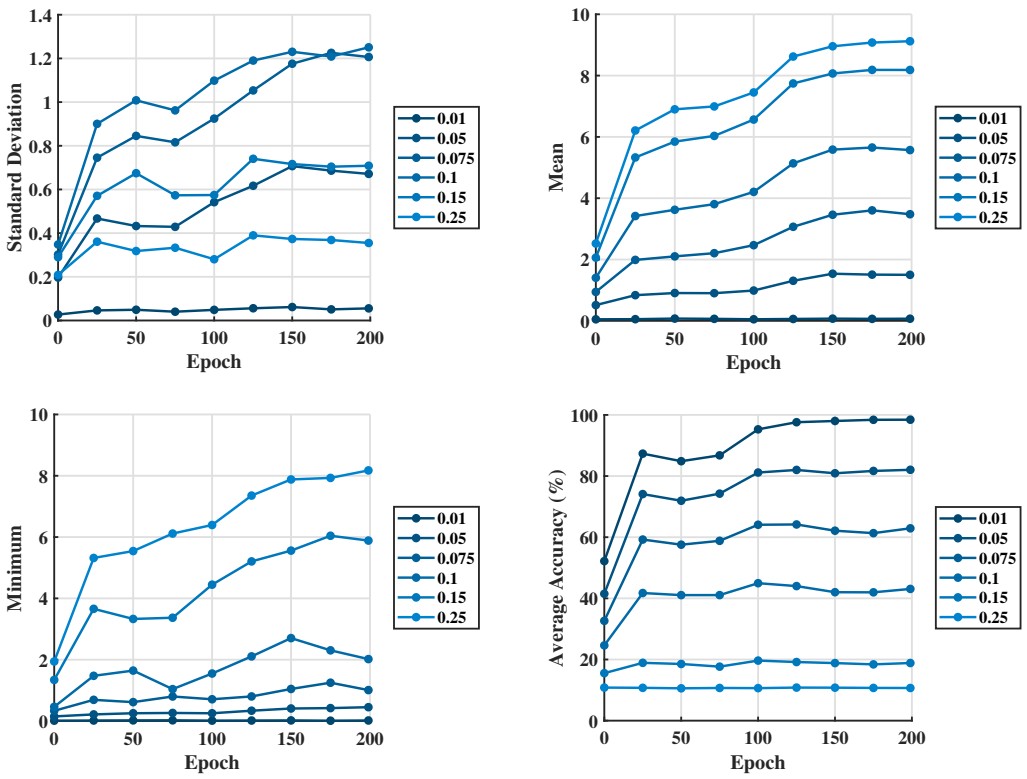

Figure 12: The weight utility of the ResNet20 (He et al., 2016) with Batch Normalization (Ioffe & Szegedy, 2015), trained by SGD with momentum and weight decay. Each legend indicates the ablation ratio.

**The longer training epochs with the advanced settings** Here, the experiments was done with the advanced settings and the larger training epochs. We used ResNet20 (He et al., 2016) with Bathch Normalization (Ioffe & Szegedy, 2015), trained by SGD with momentum ($\alpha = 0.9$) and L2 weight decay ($\lambda = 10^{-4}$). The number of training epochs is 5000. The initial learning rate of 0.1 was reduced by $10\times$ at $50\%$ and $75\%$ of the total training epoch. Other settings are the same as the experiment in Section 2.1.

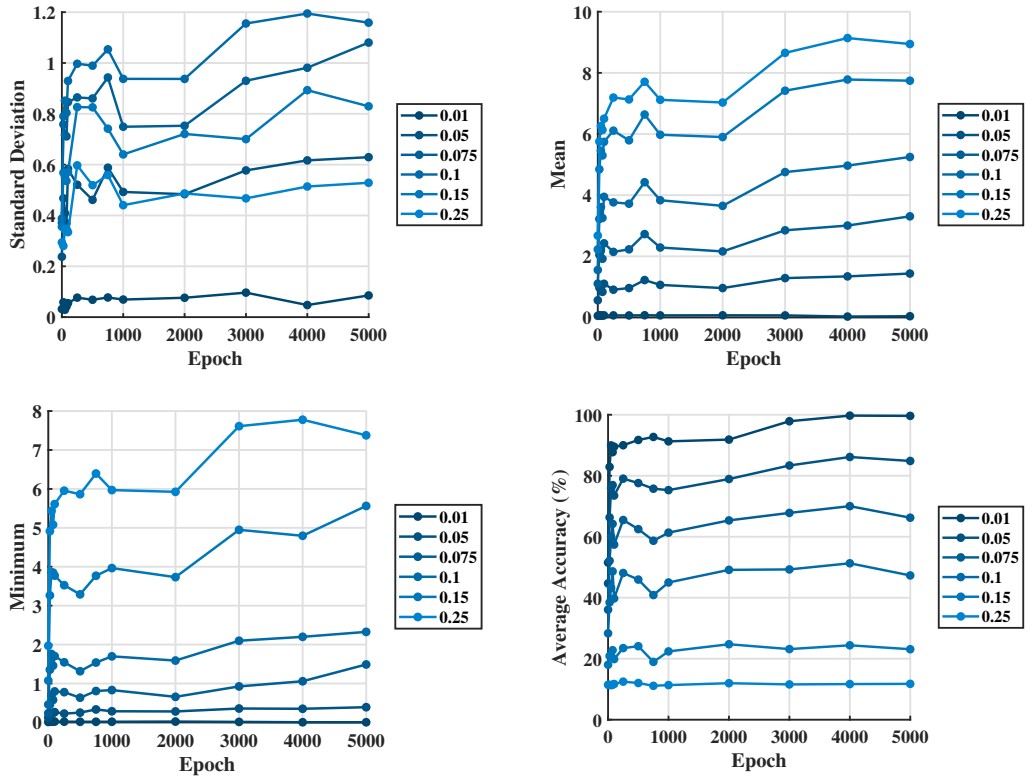

Figure 13: The weight utility of the ResNet20 (He et al., 2016) with Batch Normalization (Ioffe & Szegedy, 2015), trained by SGD with momentum and weight decay for 5000 epochs. Each legend indicates the ablation ratio.

A.3    ADDITIONAL FIGURES FOR SECTION 2.2.1

A.3.1    THE HISTOGRAMS OF THE WEIGHT UTILITY FOR THE DIFFERENT-SIZED NETWORKS.

Here, we show the histograms of the weight utility for the different-sized networks. The figures are ordered as the corresponding ablation ratio, i.e., 0.01, 0.05, 0.075, 0.1, 0.15, 0.32.

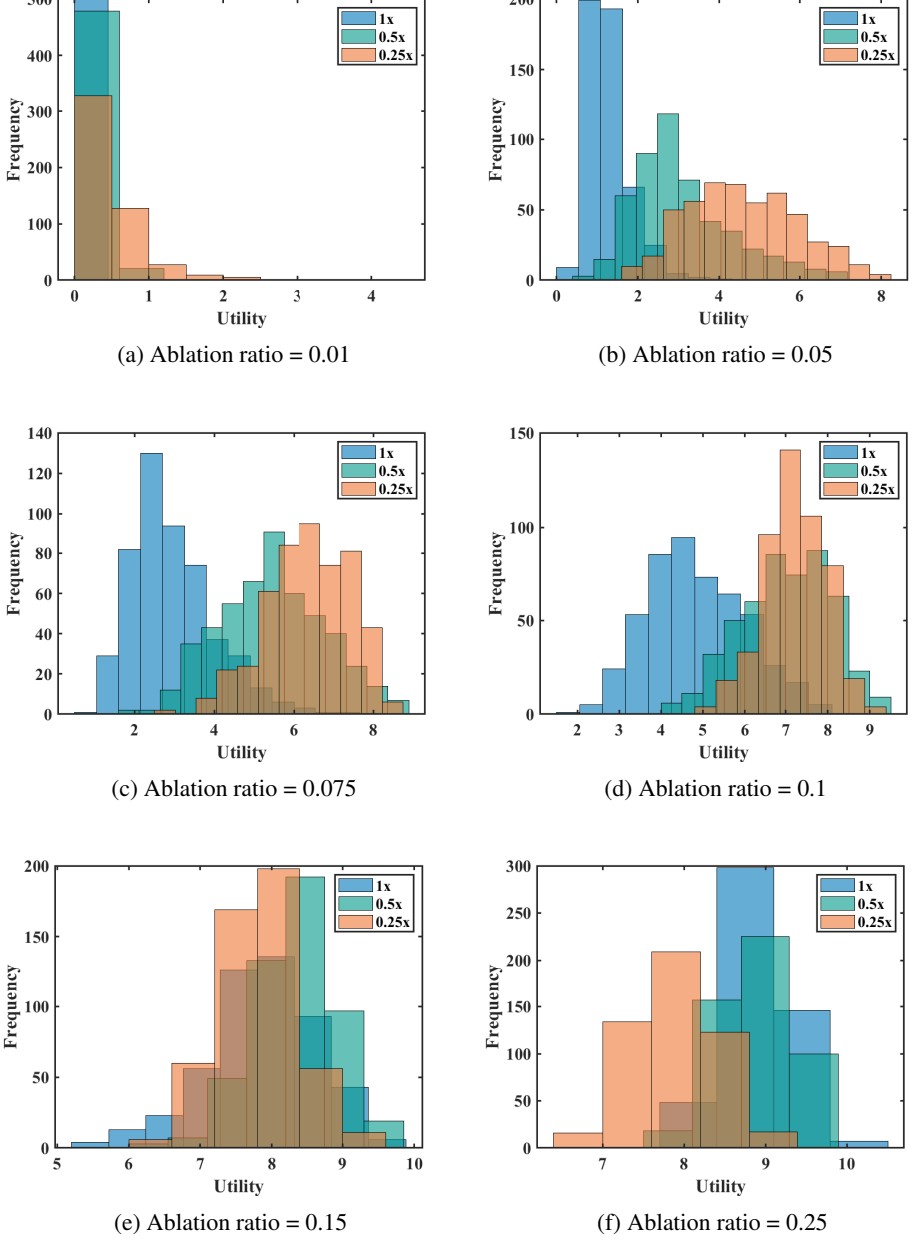

Figure 14: The histograms of the weight utility for the different-sized networks. Note that the average and the standard deviation of the weight utility are higher in the smaller network when the networks have the ability to classify. And the distributions become similar as the networks output random chances. Each legend indicates the size of the network.

## A.4 ADDITIONAL FIGURES FOR SECTION 2.2.2

### A.4.1 THE CHARACTERISTICS OF THE WEIGHT UTILITY IN THE PRUNED-AND-RETRAINED NETWORK WITH VARYING PRUNING RATIO

Here, we show the characteristics of the weight utility in the pruned-and-retrained network with varying pruning ratio.

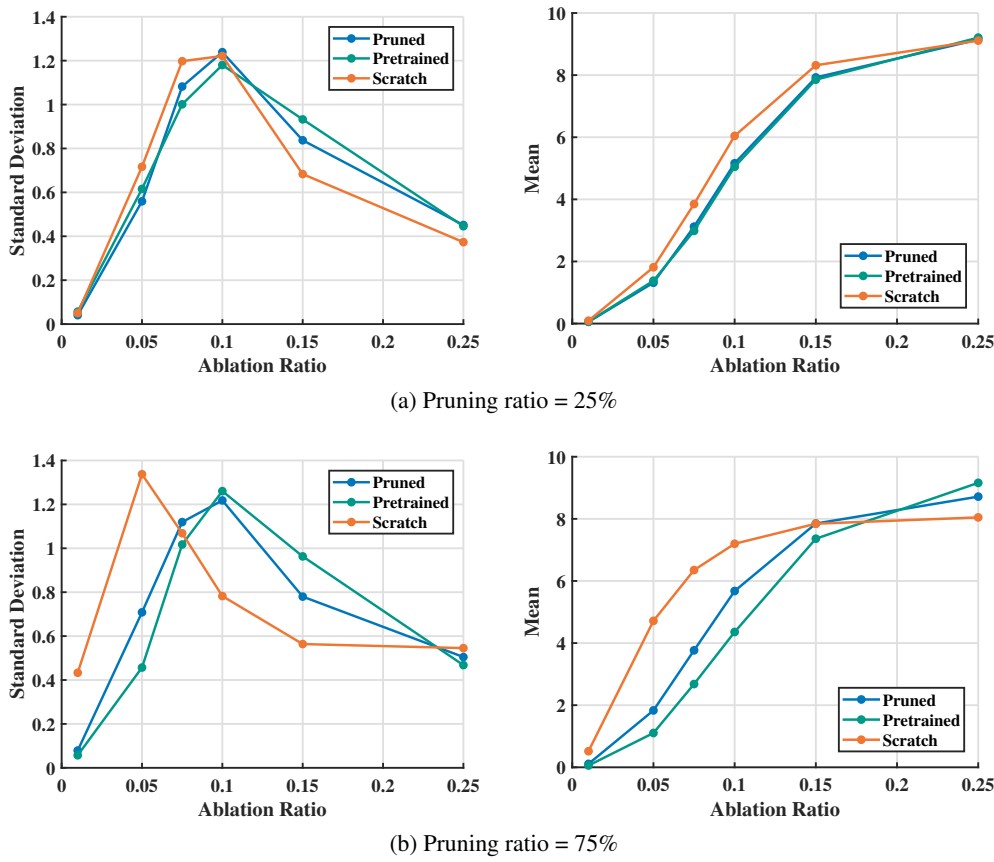

(a) Pruning ratio = 25%

(b) Pruning ratio = 75%

Figure 15: The characteristics of the weight utility in the pruned-and-retrained network with varying pruning ratio. (Top) the characteristics of the networks when the pruning ratio was 25%. (Bottom) the characteristics of the networks when the pruning ratio was 75%. Although the pruned-and-retrained network shows the similar characteristics to those of the pretrained network, it gets different as the pruning ratio is increased.

### A.4.2 THE TRAINING LOSS AND THE VALIDATION ACCURACY FOR THE EXPERIMENTS.

Here, we show the training loss and the validation accuracy for the experiments used for Figure 5 and Figure 15.

Table 2: Validation accuracy comparison. "prune" refers to the pruned-and-retrained network, and "small" refers to the small network with the same number of weights as that of the pruned network, which is trained from scratch.

| Pruned ratio (%) | Training type | Validation accuracy (%) | Train loss |
|---|---|---|---|
| 25 | prune | 92.43 | 0.008 |
| | small | 91.14 | 0.016 |
| 50 | prune | 91.84 | 0.009 |
| | small | 90.63 | 0.033 |
| 75 | prune | 91.41 | 0.026 |
| | small | 89.17 | 0.105 |

## A.5 ADDITIONAL MATERIAL FOR SECTION 3

### A.5.1 THE DIFFERENCE BETWEEN NETWORK SLIMMING AND OUR CHANNEL PRUNING METHOD

Here, we describe the difference between network slimming (Liu et al., 2017) and our channel pruning method. Most importantly, the sparsity constraint was not used for a fundamental analysis. The pruning ratio constraint for each layer was also ignored (refer to Liu et al., 2017, Section 4.5), so that our pruned-and-retrained network often collapsed when the small learning rate was used for retraining (fine-tuning, please refer to Table 2). Also, we pruned the channel by 67%, which is by far above the recommended level (refer to Liu et al., 2017, caption of Figure 1). We did not use the best hyperparameter for the method either, e.g., the scaling factors of the Batch Normalization layers (Ioffe & Szegedy, 2015), which we used the conventional value, i.e., 1, unlike the value used for the method, i.e., 0.5.

A.5.2   WEIGHT TRAJECTORY ON LOSS SURFACE OVER ITERATIVE PRUNING PROCESS

(a) Weight pruning, fine-tuning

(b) Weight pruning, rewinding

(c) Channel pruning, fine-tuning

(d) Channel pruning, rewinding

Figure 16: Loss surface visualization of each pruning method and retraining method. On each plot, $W^*, W_i^p, W_i^r$ refers to the pretrained weight, the pruned weight, and the retrained weight. The number of pruning iteration is indicated by the subscript $i$. Note the transitions in the accuracy surface over the cycles.

