# OpenReview forum: "An empirical study of a pruning mechanism"
_ICLR.cc/2021/Conference — Reject_

### Official Review · AnonReviewer2 · 2020-10-28
**Study of a pruning after training using weights' utilization measure**

**Rating:** 4
**Confidence:** 4

**Review:**

This paper addresses the following important question related to "pruning after training" framework: why, in general, smaller network trained from the scratch does not achieve the same accuracy as the pruned network (obtained from pretrained net)? For that matter, authors introduce a new mechanism ("weights' utility measure and utility imbalance") to measure a discrepancy between subnetworks (subset of weights) within each network and it seems that having a large value of that measure accounts for the networks not utilizing all of their weights.

In general, I liked the approach chosen by authors to study the effects of pretraining and weight utilization in pruning. For example, it is interesting to observe that the accuracy difference can be monotonically increasing during neural net training and that the neural networks utilization is proportion to their size. Moreover, such difference (described by utility imbalance measure) is observed independent of network size. As far as I know, such study is novel. However, I don't think that the utility imbalance measure is a unique mechanism for such analysis and one can choose other possible options. In fact, a huge variation of the neural net accuracy depending on which subnetwork (subset of weights) you choose is so obvious that there is no need to show it explicitly (fig. 1). Moreover, the core idea behind pruning is to find such best subset of weights. Therefore, I don't think that such variability can explain why smaller nets cannot achieve the same accuracy as that of pruned net.

As for the second part of the paper (loss function surface), the accuracy surface is drawn based on only THREE points, specifically the loss of the pretrained net, pruned net and retrained net after pruning. I don't think that this information is sufficient to obtain an accurate visualization. Moreover, again, I believe that the pruned-then-retrained and trained net are lying on similar surface is trivial to show since: 1) pruned-then-retrained almost always perform better than just pruning; 2) if we don't prune too much then we can expect similar performance as the original net. Therefore, I don't see much contribution here.

Minor concerns: \
 . some parts of the paper is relatively hard to follow (e.g. in Section 2, jumping from experiment setting to discussing results and vice versa);\
 . colors in some figures (e.g. fig. 1 and fig. 2) are really hard to distinguish since they are very similar.\
 . section 2.2.2 describes utility imbalance in a pruned network (before or after retraining). I'm just curious what would be that measure for iterative pruning schemes. For example, as in (Han et al., 2015) and [1,2].

[1] Zhang et al. Adam-ADMM: A unified, systematic framework of structured weight pruning for DNNs. 2018.\
[2] Carreira-Perpin˜an, M. and Y. Idelbayev. Learning-compression algorithms for neural net pruning. 2018.

---

> ### Author Response · Authors · 2020-11-13
> **Answers for the review**
>
> We appreciate your genuine review. Here are some opinions and answers for your concerns and questions:
> 1.	Regarding ‘… I don't think that such variability can explain why smaller nets cannot achieve the same accuracy as that of pruned net …’: We appreciate your point. If there is a utility imbalance among the weights in a network, it means some of the weights are less utilized. However, when a network utilizes the larger portion of the weights, as the pruned-and-retrained network does in 2.2.2, it achieved better performance. However, we see that the definition of the weight utility does not directly remind of what we described above. We will consider that. We appreciate for the point once again.
> 2.	Regarding ‘…the accuracy surface is drawn based on only THREE points…’: The study of drawing loss surface is commonly done with 2~3 points [1,2,3]. However, we also see your point.
> 3.	Regarding ‘… pruned-then-retrained and trained net are lying on similar surface is trivial to show since …’: We conducted this study due to [4]. We were curious whether the pruned-and-retrained network stays in the basin of the pretrained network, even if we retrain the pruned network with large learning rate. The pruned network might have escaped from the original basin and the retrained network could have gone to the other basin with the comparable accuracy. However, from the visualization, we could see that the pretrained network and the pruned-and-retrained network shares the same basin, as the two points connected in any dimension are already connected no matter what happens in other dimensions. On the other hand, with drastic pruning, it is probable that the two networks are in the different basins, as we cannot guarantee the same thing happened in other dimensions.
> 4.	Regarding ‘… I'm just curious what would be that measure for iterative pruning schemes.…’: For weight pruning, we used [5]; and for channel pruning we used the criterion in [6]. To avoid misunderstanding, they were described in Experiment details subsection in Section 5.
> 5.	We appreciate the other advices and opinions as well.
>
> [1] Garipov, Timur, et al. "Loss surfaces, mode connectivity, and fast ensembling of dnns." Advances in Neural Information Processing Systems. 2018.
>
> [2] Li, Hao, et al. "Visualizing the loss landscape of neural nets." Advances in Neural Information Processing Systems. 2018.
>
> [3] Evci, Utku, et al. "The Difficulty of Training Sparse Neural Networks." (2019).
>
> [4] Renda, Alex, Jonathan Frankle, and Michael Carbin. "Comparing Rewinding and Fine-tuning in Neural Network Pruning." International Conference on Learning Representations. 2019.
>
> [5] Han, Song, et al. "Learning both weights and connections for efficient neural network." Advances in neural information processing systems. 2015.
>
> [6] Liu, Zhuang, et al. "Learning efficient convolutional networks through network slimming." Proceedings of the IEEE International Conference on Computer Vision. 2017.

---

### Official Review · AnonReviewer4 · 2020-10-28
**Interesting direction but conclusions are not supported by the selected measurements**

**Rating:** 4
**Confidence:** 4

**Review:**

Summary: This paper dives into why small pruned networks don’t train as well as large networks. They come up with a measure of weight utilization and claim that networks of all sizes only use a portion of their weights during training, and the imbalance increases during optimization. Additionally, they visualize the accuracy surface on the plane defined by the pretrained, pruned, and retrained networks and find that the retrained networks end up in the same basin as the pretrained networks.


Overall: The paper is going in an interesting direction, but I find the paper to be unclear and I do not see how their chosen measurements lead to their claims. Thus, I do not recommend acceptance.


Pros:
* It is important to understand the mechanisms of how pruning works, as well as the role of overparameterization in neural networks.
* It is also interesting to see how the ratio of weights used change with respect to network size or sparsity.
* Visualizing the loss/accuracy landscape between the three networks (pretrained, pruned, pruned+retrained) is interesting and I have not seen that before.

Cons:
* The need for overparameterization to achieve good performance has been studied before (https://arxiv.org/abs/1805.12076, https://arxiv.org/abs/1804.08838), though I do see the benefit of analyzing this from a different angle.
* There are also studies of how neural networks do not use all of their weights during training (https://arxiv.org/abs/1909.01440)
* The given utility measure is not the best way to measure how useful certain weights were. Some weights could have been very important for moving around during training, even if they end up unimportant or at a low magnitude. For instance, overparameterization provides more degrees of freedom that may be necessary in optimization (e.g. as discussed in https://arxiv.org/abs/1906.10732). You could include additional measures such as LCA (from https://arxiv.org/abs/1909.01440) to measure how “useful” certain weights were throughout training.
* Suggestion: rather than choosing random subsets of weights to prune/ablate in definition 3, why not prune based off of some existing measure? E.g. prune by magnitude, using the n% lowest for $W_i$ (what magnitude pruning would choose) and perhaps the n% highest for $W_j$ for comparison. Your current measure won’t be able to pinpoint if the network heavily relies on, say, a specific 10% of its weights, because pruning a random subset of weights means that you prune both the important weights and the unimportant weights. Currently, this measure only evaluates the network’s sensitivity to random pruning.
* Section 3 results: while it is useful to visualize the accuracy surface in 2D, a 2D picture does not tell the full story for high dimensional spaces: the existence of a connector in one particular dimension does not mean that two points are necessarily in the same basin. Further, if the pretrained model and the pruned+retrained model are in the same basin, that would mean the pruned+retrained model achieves the exact same accuracy as the full model, which is not the case unless the sparsity is low. Perhaps it would be interesting to see these visualizations at different pruning levels, where accuracy begins to drop. Also, why use accuracy rather than loss?

There are also several points in the writing that are unclear. While I do not want to penalize the paper for writing style, these issues make it difficult to understand the ideas in the paper. If it is just my misunderstanding, I will gladly read any explanations that the authors can provide.
* Definition 3: don’t you need a $\delta$ for utility imbalance? How do you convert standard deviation to utility imbalance? How do you “show that the utility imbalance among the weights exist” if utility imbalance is not a binary measure?
* Please give a more intuitive description of utility and utility imbalance. What does it mean for networks to have high utility imbalance given the definition you provided? I don’t agree that having a large standard deviation in utility of the random $W_i$ signifies that the neural network utilizes the weights unevenly; the implication I see is that the network’s output is very sensitive to which random subset of weights you prune.
* Figure 1 left: why would a drop in accuracy after removing weights imply the utility imbalance? Right: what does “ablation regarding the magnitude of weights” mean? How do you choose which weights to ablate?
* Section 2.1.1: this explains why magnitude pruning works well but does not explain how initialization causes utility imbalance.
* Section 2.1.2: how is $|\Delta\delta|$ the “amount of ablation”? How can $\delta$ be added to $W$ if $\delta$ is used in $|U(W_i) - U(W_j)| \geq \delta$? Also, my guess would be that utility increases over training because at the beginning of training, the network is close to random, so removing some subset of weights won’t change it as much (can’t get much worse than random) as ablating a more trained network. Thus it is only natural that it increases - I don’t see how this supports the claim that the loss landscape became sharper. Also, what does “each of the weights is struggling to be utilized more” mean?
* Section 3 paragraph 3: what are the “assumptions for the heuristic”?

---

> ### Author Response · Authors · 2020-11-13
> **(Continued from above)**
>
> (Question 1, 2) Here are intuitive explanations for the definitions. We should have described it in the paper. We are sorry if it made you confused and distressed.
>
> (Weight utility, def1) Let’s say there is a trained network. It has various weights with various importance. And then we ablate a weight. If we ablate an important weight, the outputs of the original network and the ablated network will differ a lot. In this case we say the weight is highly utilized. On the other hand, if the weight is less important, the outputs will vary less and we say the weight is less utilized by the network. Since ablating a single weight merely affects network, we extend this concept to a set of weights.
>
> (Definition of utility imbalance, def2)
> If two weight sets have different utility, or importance, we can say that there is utility imbalance between the sets.
>
> (Measure of utility imbalance, def3)
> Let’s say there are some weight sets we draw from a network. And we measure the utility of each weight set. If the utility of the weight sets differs from each other, then we can say there is utility imbalance between the sets. Likewise, we can measure how much the utility values of the sets deviate from each other. Intuitively, one can measure variance among the utility of the sets, where we actually used standard deviation.
>
>  (Q 3) About Figure 1. (Left) Let’s say we have two weight sets A and B with the same size in a network. And we get 80% and 20% accuracies when ablating set A and B from the network respectively. Then we can say the network relies more on the weights in set B than those in A, and thus we can also say there is utility imbalance between A and B. Likewise, the accuracies in the Figure 1 variate even when ablated by the same portion (as is represented by min-max bar). For example, when we ablate 10% of the weights, the accuracies vary from about 20~80% in the left figure. (Right) We obtained the figure by following procedure: the weights were sorted by their magnitudes and chunked into parts by the given ratio. And each chunk was ablated sequentially. For example, after sorting the weights by their magnitude, we divide them into 10 chunks, which corresponds to 0.1 in the legend, and ablated each of them to get each point in the figure. Each value in the magnitude axis represents the maximum magnitude of the weight in each chunk.
>
> (Q 4) Lottery Ticket Hypothesis (LTH, Frankle & Carbin, 2018) states that there is a subnetwork from initialization that can achieve similar performance to that of the original network given the similar training time. If there is such a subnetwork, then the other weights in the network seem to have small effect on training the whole network. However, we recognized that there is a logical leap. There can be a counter example like the case you suggested earlier (https://arxiv.org/abs/1906.10732). We should provide more evidence. Thank you for pointing out.
>
> (Q 5) The \(\delta\) in 2.1.2 is just an expression for weight change. It is unrelated to the one in the definition. Also, we appreciate your great point about KL-divergence. Other measurements such as Euclidean distance between the logits may be more proper than probability-based measure to assert our claims. On the other hand, we described as ‘each of the weights is struggling to be utilized more’ to indicate that the utility of each weight keeps increasing, rather than the utility of some weights are suppressed to go to a better minimum for a greater good.
>
> (Q 6) Okay. Just ‘heuristic’ can be better. Thank you.

---

> ### Author Response · Authors · 2020-11-13
> **Answers for the review**
>
> We appreciate your generous and sincere review. We are grateful to the pros in the review. Here are the answers for the cons and the questions:
>
> (Cons 1,2) Thank you for recommending genuine papers. We will refer to those.
>
> (Con 3) We measured the weight utility and the utility imbalance throughout training in 2.1.2. Please let us know if this is not what you mean.
>
> (Con 4) The random ablation was chosen for the following reasons. 1) If the measurement were based on a certain criterion, we cannot tell whether a result of an experiment is due to the importance of the weight or the characteristic of the criterion. For example, if we set the magnitude of weight as the criterion and acquire a result from an experiment, then we cannot be sure whether the result came out because the weight is important or the weight has large magnitude. 2) Also, since there is no absolute criterion for the weight importance, there is a limitation for a criterion to represent it. For example, it is quite well-known that the magnitude of weight is highly related to the importance of weight (Han, 2015), however, there are also some weights with small magnitude but still important. On the other hand, ablation is a direct way of measuring the weight importance, although it can only measure as a group and show overall trend when sampled repetitively.
>
> (Con 5)
> 1) Answer for ‘… the existence of a connector in one particular dimension does not mean that two points are necessarily in the same basin.’: This is a very good point. We described as those since if two points are connected in any dimension, then it means they are already connected no matter what happens in other dimension. On contrary, if we observe two points being in the different basins, we can only say that there is a possibility that they are separated, since they may be connected in other dimensions as you depicted.
>
> 2) Answer for ‘… if the pretrained model and the pruned+retrained model are in the same basin, that would mean the pruned+retrained model achieves the exact same accuracy as the full model…’: Even if the two models are in the same basin, they can have different accuracy. This is because one basin consists of a part of continuously-varying loss surface.
>
> 3) Answer for ‘why use accuracy rather than loss?’: With accuracy surface, we can intuitively know how abruptly the loss surface changes. There is no particular reason other than that. Actually, the loss surface looked better 😊

---

> > ### Comment · AnonReviewer4 · 2020-11-23
> > **Thank you for the clarifications**
> >
> > Some additional comments/questions:
> >
> > "We measured the weight utility and the utility imbalance throughout training in 2.1.2." --> I agree that you evaluated the utility imbalance measure at different points in training, however, the measure at each point only captures which weights are useful at that exact moment. For instance, the utility imbalance measure at epoch 100 can answer the question "which weights at epoch 100 will harm the network if ablated right now" but not "which weights' movements were important in getting the model from random initialization at epoch 0 to the better performing model at epoch 100."
> >
> > "If the measurement were based on a certain criterion, we cannot tell whether a result of an experiment is due to the importance of the weight or the characteristic of the criterion" --> This is a good point. However, I'm still not convinced that random ablations can measure the imbalance between important and unimportant weights. Let's say a network as 100k weights, and only 10k of them were useful (for simplicity, let's say "useful" is binary). If you ablate a random 10% of the weights, you will ablate an expected number of 1000 useful weights and 9000 unimportant weights for each set. There will be some variance between different random sets, but since each set will very likely ablate both useful and unimportant weights, it cannot capture the fact that only 10% of the weights were useful.

---

> > > ### Author Response · Authors · 2020-11-24
> > > **Thank you for the discussion**
> > >
> > > Here is are answers for the questions. Rather than which weight finally becomes important or an exact portion of important weights, our interest is on 1) average importance of weights and 2) variance among importance of weights during/after training. Therefore, we measured how critical the weights become (on average) and how different importance among the weights become (variance) during training, regardless of the state after training.
> > >
> > > In addition, we regard random ablation can measure the imbalance of the importance among weights. For example, let's assume we have two networks, network A and B, and each network achieves 100% training accuracy. And we randomly ablate 10% of the weights in each network and measure the training accuracy respectively. The experiment is repeated for hundreds of times. And then the ablated networks from network A achieve 60% accuracy on average, and so do the networks from network B. However, the importance imbalance among the weights in the two networks can be different. If the randomly ablated networks from network A achieve 30\~90% accuracy and the networks from network B achieve 59\~61% accuracy, we can say that the importance of the weights is more imbalanced in network A. In other words, it can be inferred that network A relies on a few strong weights, so that it achieves 30% accuracy when the important ones are gone and 90% accuracy when they are retained. On the other hand, network B consists of the weights with the similar influence. In this way, we don't measure the portion of the important weights, but the deviation of importance among the weight sets. The ablation ratio has nothing to do with the ratio of important weights.
> > >
> > > Thank you.

---

### Official Review · AnonReviewer3 · 2020-10-29
**The assumption does not always hold; imbalance is not surprising; the study on imbalance is quite limited**

**Rating:** 4
**Confidence:** 4

**Review:**

The paper proposes to answer the question why "a network with the same number of weights as that of the pruned network cannot achieve similar performance when trained from scratch". Then it proposes an hypothesis that the small model "does not utilize all of its weights either". To prove this hypothesis, it goes on to define and study the "utility imbalance" of the weights and its changing with the pretraining, pruning, etc. Some visualization analysis was provided too.

I appreciate the detail empirical studies conducted in this paper, but I have some serious concerns:

1. The paper cites [1] for the assumption that "a network with the same number of weights as that of the pruned network cannot achieve similar performance when trained from scratch". This is the basis for the whole study of the paper. This assumption seems to be in contradiction with conclusion in [2] for some cases, including the experimental setting this paper studies. In fact, it is well documented both in [1] and [2] that a network trained from scratch can reach a similar performance as a pruned-retrained network, under relatively large learning rates, in the case of unstructured pruning for CIFAR-10, or any structured pruning. The learning rate used is 0.1 for the CIFAR-10, which this paper also uses. This basically makes the assumption of this study (and thus the goal of proving the hypothesis) at least not valid in some cases. I am aware that this issue on learning rate was mentioned in section 2.1.1, but that still doesn't address the basic assumption and aimed goal of this study.

2. The paper demonstrated there is imbalance in the network weights' "utility", but did not try to investigate which part of the weights are of higher utility. I think this is a more important problem than its relation with network size. For example, in [2], the authors showed the imbalance usage within 3x3 convolutional kernels.

3. The utility imbalance is not really a surprising phenomenon. After or during training, only with these utility imbalance we can prune network according to various criteria, and beat random pruning. Thus this should not be surprising to the literature. For imbalance at initialization, the paper directly cites [1] without much further study.

4. The paper only discusses the conclusion at [1] for the utility imbalance at initialization. I think some experiments could help better understand the cause of imbalance at initialization. For example, do shallower layer's weights on average has larger utility? or those weights with large magnitudes? I wouldn't be surprised if the imbalance can be explained by the differences with initialized weight magnitudes. So again the question is how much of the imbalance is explained by the lottery hypothesis, and how much is due to other reasons. This goes back to point 2, where my concern is the paper did not investigate which weights are of higher utility, either at initialization, during optimization or after.

5. The paper title is a bit too vague. The conclusions are kind of vague too. The conclusion section mentioned a lot of content was studied/discussed/defined/investigated, but didn't give what the findings are. This is also a reflection that the paper/study itself didn't have a strong conclusion either.

In total, there are some interesting analysis on the changing process of the defined imbalance. However, the paper seems to lack a central conclusion that can help future practices, or help people gain deeper understanding. Network weight imbalance is expected, not surprising. The paper did not dig further with the imbalance by investigating which subsets of weights are more useful, and did not give much explanation with experiments on why the imbalance exists. Last but not least, the assumption and the attempted hypothesis does not always hold in the first place, in particular even in paper's experimental setting (large lr).

[1] The Lottery Ticket Hypothesis: Finding Sparse, Trainable Neural Networks [2] Rethinking the Value of Network Pruning


I appreciate the author response but unfortunately they are still a bit vague (1,3), or not supported with experiments (2,4). I still maintain my rating of 4.

---

> ### Author Response · Authors · 2020-11-11
> **The answers for the concerns**
>
> We appreciate your earnest review. These are our answers to your concerns:
> 1.	We are sorry for the confusion caused by using the expression ‘cannot’. The assumption we made is based on the most basic settings for training a neural network. There are some cases that small network trained without pretraining can achieve the accuracy similar to that of large network, as you indicated. However, those cases need special conditions, i.e., for [1], one needs to know which weights belong to the winning tickets in advance; and for [2], one needs to train small network much longer to achieve the accuracy of large, unpruned networks. Our assumption rather considers the case where small network and large network are trained under the same conditions, i.e. for [1], the unpruned network vs. pruned and randomly reinitialized network; and for [2], the unpruned network vs. Scratch-E cases, where the accuracies of the unpruned network were generally higher than those of Scratch-E. However, thanks to your advice, we recognized that our assumption needs to be clarified.
> 2.	The problem of finding which weight is important, or more utilized, is a fundamental question in this area. When regarding the important set of weights, it becomes NP-hard and when regarding weights individually, there are some characteristics related to the importance, e.g., magnitude of weight, but they are only sufficient conditions and might be never fulfilled. We agree that the subject is important and should continue to be studied despite the difficulties.
> 3.	The utility imbalance is already well-known and is not a surprising phenomenon. What we tried to convey in this paper was the cause of the utility imbalance and the characteristics of it.
> 4.	Thank you for the excellent idea, ‘how much of the imbalance is explained by the lottery hypothesis, and how much is due to other reasons.’ We will study it.
> 5.	The choice of the words was not intended. It happened since we are non-English speakers. Our findings are described in the lists of contributions in 1. Introduction.

---

### Official Review · AnonReviewer1 · 2020-11-02
**Inscrutable Claims and Inscrutable Methods**

**Rating:** 4
**Confidence:** 2

**Review:**

# After Rebuttal

I have read the response to my review and the responses to the other reviews. The summaries of the paper in the other reviews helped to clarify my understanding of the research question the authors were aiming to answer and how they went about doing so. I have re-read the paper and the revisions have helped to make this story clearer.

With that said, I remain concerned with many technical aspects of the paper, for example:
* The scale of the networks studied.
* I still don't understand why this particular definition of utility imbalance is well-motivated.
* I generally don't think that two dimensional loss landscape visualizations are informative since they discard an enormous amount of information from the full loss landscape. To show that minima are related, I think it is better to use interpolation (mode connectivity).

I am also still concerned about the writing. The revisions, alongside the other reviews, were enough for me to get a sense for the research question and technical story, but I still struggled to make sense of the details.

Since the other reviewers appear to have better understood the paper, I have raised my score to a 4 and decreased my certainty to a 2. I suggest that the AC weight my review much less heavily than the other reviewers, who seem to have better understood the technical details.

# Overall

This paper is inscrutable. The research question is unclear and the hypothesis is vague and imprecise. The metric being examined ("utility imbalance") is never justified in name or in definition, and broad, sweeping, unsubstantiated claims are made that it has to do with how the network uses its weights or the sharpness of the minima (among other aspects of deep learning). Section 3 seems completely unrelated to the previous analysis, and I don't understand how Sections 2 and 3 relate to form a cohesive story. I can't make sense of this paper - its question, its claims, or its method - and I'm an expert reader on topics of pruning and lottery tickets. I can't imagine what a non-expert reader would be able to glean from this paper.

# Score

I therefore strongly recommend rejection (score of 2).

# To Improve My Score

To Improve my score, the authors need to:
* Explain what the research question is, precisely.
* Explain what the hypothesis is, precisely.
* Explain (in detail) what the justification is for "utility imbalance" and why it corresponds to meaningful behaviors of a neural network. The authors will need to include evidence that this corresponds to the claimed attributes of a neural network, which will include a precise definition of what it means for a network to "utilize" a weight and how this metric measures that.
* Explain why this is called "utility imbalance"
* Explain jhow the results in sections 2 and 3 related to each other and the larger narrative/takeaway that they provide.
* Explain why the network under study changes mid-way through the paper. Is there a reason for this? When I see this in papers, it's usually because the authors tried this on both networks and are only showing the positive results and are hiding bad results; to assuage this concern, the authors should show all results for both networks.

From there, I'll need to re-read the experiments to see if I can make sense of them. In its current state, I do not believe that this paper makes a contribution to the scientific literature.

# Notes

## Abstract

What do the authors mean by "the pruning mechanism"?

## Intro

What prior work has claimed that "the reason for training the large network [is] to obtain a good minimum"? I don't recall having previously seen this specific claim, nor the implied negation of this claim: that training a pruned network finds a "bad minimum." What makes a minimum "good" or "bad" and how can we measure this?

"Cannot achieve similar performance when trained from scratch" - this only occurs with a new random initialization, as Frankle & Carbin 2018 show (at least for the small-scale networks they examine).

What do you mean by a "utility imbalance"? Which networks aren't "utilizing all their weights," the unpruned networks or the pruned networks?

After reading the introduction, I really have no idea what the paper is about. I don't understand the research question, the main hypothesis, the methodology, or the findings. the authors should make efforts to clarify this story. I'm confused, and that means I'm going to have a hard time making sense of the rest of the paper.

I also can't make sense of Figure 1. I don't know which network this refers to, and I still don't know what a "utility imbalance" is.

## Section 2

Why is there an assumption that a large network "does not utilize all its weights and thus can easily be compressed?" There are many possible reasons that one can prune a network, and we often find in the literature that pruning reduces accuracy. The entire point of retraining is to recover this accuracy. But since accuracy went down, it seems that these weights did serve some purpose.

Is N_small the same pruned architecture, or is it a different pruned architecture, a smaller dense network, etc.?

What does it mean for a network to "utilize all its weights"?

In Definition 1, does the pruned network include retraining?

Why is this measure of utility imbalance meaningful or useful? Networks whose outputs have very different distributions may get similar accuracy, so KL divergence doesn't seem to be esxpecially meaningful here. I also don't see what this has to do with utility.

#### Section 2.1.1

I don't follow this logic, and I'm an exceedingly well-informed reader when it comes to lottery tickets. What is the purpose of this section? Is utility being used in a formal way or an informal way? If formal, I don't see a proof or derivation to support the claims here. This seems like an argument without any evidence.

#### Section 2.1.2

It is unacceptable that the name and details of the network used for this paper are buried in an appendix. The network in question, a small convolutional network, appears to be similar to the small convolutional networks used by Frankle & Carbin. These networks are not representative of larger-scale settings for lottery ticket behavior see ("Linear Mode Connectivity and the Lottery Ticket Hypothesis" (Frankle et al., 2020)), and the results in this paper should not be taken to be general.

The first paragraph of 2.1.2 suggests that the "utility imbalance" metric has anything to do with the specified claim made by Frankle & Carbin, and I see no reason to believe that.

I don't know how to interpret the "utility imbalance." It has a fancy name and it has a fancy definition, but I have no idea what it means. Graphs of the utility imbalance therefore aren't very enlightening.

Why does the utility have anything to do with the network "utilizing the weights more effectively"?

What does the utility imbalance have to do with the sharpness of the loss landscape?

Why are the weights and SGD being anthropomorphized? e.g., "The weights is struggling to be utilized more, rather than SGD purposely differ the utility among the weights?"

I'm completely lost in this section, as the above comments hopefully convey. I have no idea what is being measured, why it is being measured, or why the results have anything to do with the claims. These experiments are also being conducted on a single, non-standard network, so I don't understand why these results should generally be true.

## Section 2.2

What does it mean to "utilize a weight"?

Why were the weights rescaled? I don't understand what "This was to control the number of feature maps and thus avoid any architectural bias." means.

Why did the experiment suddenly switch networks from what was being used in 2.1?

I made a good-faith effort to read the rest of the section from here, but I wasn't able to make much sense of why the evidence substantiated the claims. Please see my general comment at the beginning of the paper.

## Section 3

"Empirically, the generalization gap is known to be closely related to the geometry of the loss basin." This is hotly contested in the literature. I'm not sure you can say this so strongly.

Figure 6 needs a legend to explain what the networks are. The text never explains what W*, Wp, and Wr are.

I have no idea how this section relates to the rest of the paper.

---

> ### Author Response · Authors · 2020-11-13
> **Answers for the review**
>
> Okay. I am the first author of this paper. As I wrote the part you have trouble with, I will answer to your questions.
>
> (Q1, 5) We tried to validate the commonly used method, i.e. pretraining a large network and pruning and retraining it. We investigated the method in two parts: pretraining part and pruning-and-retraining part. Each part was studied with distinct measurement/method.
> (Q2, 3, 4) I see your point and please refer to the intuitive explanations for the definitions we provided for AnonReviewer4, i.e. (Weight utility, def1), (Definition of utility imbalance, def2), (Measure of utility imbalance, def3). Also, we recommend you to have a look at the first answer we provided for AnonReviewer2.
> (Q6) The networks and the optimizers used for the entire experiments were the same. We only provided the additional setting in 2.1.2, since we needed to show that the characteristics of the weight utility solely come from SGD, not from skip connection, Batch Normalization, momentum nor weight decay. Moreover, the experiments in 2.1.2 were even done with the setting which is the same as that of all the others, and the corresponding results were provided in Appendix A.2.4, ‘An experiment with the advanced settings’. We recommend you to check it.
>
> I am deeply sorry that you had such a hard time reading this paper. I see how it happened and hope I do better next time. However, it is unfair to raise an ethical issue. Even if it came from a misunderstanding, you could have just asked us to conduct additional experiments. Nonetheless, I do appreciate your hard work and good points you made. Thank you

---

### Decision · Program_Chairs · 2021-01-07
**Final Decision**

**Decision:**

Reject

**Comment:**

Pruning is an important problem in practice. The angle of this study is also interesting. The key concept proposed by this submission is called the "utility imbalance" of the weights.  There are many concerns raised by the reviewers. Let us summarize some of them here: (1) hard to follow even for the domain experts; (2) the definition and motivation on "utility imbalance" are unclear ; (3) loss landscape visualizations are too much simplified to be informative.  There are also lots of concerns on writings. The rebuttal did help clarifying some details. However, most of the concerns still remain. We hope the detailed comments from the reviewers will be useful for the authors to polish this work.